# Effects of Deposition Pressure on the Microstructural and Tribological Properties of CrAgCeN Coatings Prepared by Magnetron Sputtering

**DOI:** 10.3390/ma16031141

**Published:** 2023-01-29

**Authors:** Wei-hang Chang, Hai-chao Cai, Yu-jun Xue, Xian-qing Lei, Hang Li

**Affiliations:** 1School of Mechatronics Engineering, Henan University of Science and Technology, Luoyang 471003, China; 2Henan Key Laboratory for Machinery Design and Transmission System, Luoyang 471003, China; 3Nanyang Institute of Technology, Nanyang 473004, China; 4Longmen Laboratory, Luoyang 471000, China; 5State Key Laboratory of Aviation Precision Bearing, Luoyang LYC Bearing Corporation, Luoyang 471023, China

**Keywords:** deposition pressure, magnetron sputtering, rare earth, CrAgCeN coating, microstructure, mechanical properties, friction coefficient, wear-resistant

## Abstract

This study investigates the effect of deposition pressure on the microstructure and tribological properties of CrAgCeN coatings synthesized via unbalanced magnetron sputtering. The CrAgCeN coatings presented a face-centered cubic structure. As the deposition pressure increased, the surface grain topography of the CrAgCeN coatings transformed from a looser pyramidal structure to a denser structure, while their hardness *H* and elastic modulus *E* first increased and then decreased. The strengthening effect was mainly attributable to Ag and Ce elements. Conversely, the coefficient of friction (COF) and wear rates of the coatings reduced and then increased. Under 0.6-Pa deposition pressure, the COF and wear rate of the CrAgCeN coating were minimized (0.391 and 3.2 × 10^−7^ mm^3^/(N·m), respectively) while the *H* and *E* were maximized (14.2 and 206.2 GPa, respectively). The values of hardness, wear resistance, resistance of elastic strain to failure (*H*/*E*) and resistance to plastic deformation (*H*^3^/*E*^2^) were improved for the coatings by Ce. The wear mechanisms were adhesion and delamination. The wear mechanisms were adhesion and delamination. Selecting the appropriate deposition pressure can improve the tribological properties of the CrAgCeN coatings. The received results of research in this study allow us to establish a rational coating composition for deposition on tools providing an increase in machining efficiency of the materials used in engineering. CrAgCeN coating with excellent properties may be applied to steel substrate through the combined action of corrosion, high temperature and mechanics.

## 1. Introduction

The performances of tools, dies and molds in various applications are commonly improved by depositing a several-micron-thick hard coating [1,2]. Transition metal nitride coatings, mainly based on titanium and chromium, have become essential in modern wear and corrosion technologies [3,4]. As a coating, CrN exhibits stronger wear-resistance properties than TiN because its friction coefficient is slightly lower than that of TiN [5,6]. CrN films have been widely used in the industry owing to their high hardness and toughness [7,8,9], good oxidation [10] and corrosion [11] resistance, prominent tribological properties [12,13], and excellent adhesion to substrates [14]. However, most of the transition metal nitride coatings have a high friction coefficient (0.6–0.8) and their tribological effectiveness [15] is insufficient under severe tribological conditions such as high temperature, high-speed operation and high-applied pressure. To meet these demands, researchers have developed extended systems based on these coatings. For instance, a third element, such as Ti, Al, Nb, Hf or Ag, has been added to the traditional binary nitride film to improve its mechanical and tribological properties [16,17,18].

To reduce friction, a CrN surface can be deposited with noble metal atoms. For example, Ag has been introduced as a lubrication phase into a hard CrN matrix. Ag is a soft metal that reduces the friction of hard surfaces over a wide temperature range (from room temperature to 700 °C) [6,19]. Therefore, when added as a solid lubricant, it improves the tribological properties of the coating [20,21,22,23]. However, as a solid lubricating phase, the Ag element will reduce the hardness of the coating, and the rapid depletion of the Ag phase will cause a lower wear life of the CrN/Ag coating, which is not conducive to ensuring the service performance and life of the workpiece. Rare earth (RE) elements have been called the ‘vitamins’ of metals, meaning that a trace amount of RE can greatly enhance a metal’s mechanical properties [24,25,26]. RE additives have been widely used in surface technology. Doping with small amounts of Ce can improve the microstructure, surface hardness, wear resistance, bonding strength, high-temperature oxidation resistance and corrosion resistance of a coating [27,28]. Rare earth elements have special physical and chemical properties and have been widely used in surface engineering. Nevertheless, there are few studies on CrN introducing Ce by magnetron sputtering. Thus far, coupling the deposition of noble metal atoms and the doping of RE elements to co-modify CrN has been rarely reported.

In the present paper, an Ag and Ce (Ce-Ag) co-modified CrN coating was prepared using the closed-field unbalanced magnetron sputtering method and its tribological properties were investigated. The advantages of magnetron sputtering are high deposition rate, uniformity of the thin film, low operation temperature and flexibility of the process parameters (substrate temperature, working pressure, target-to-substrate distance, substrate bias, target power and deposition time) [7,8]. The CrN coating performance is strongly affected by the deposition parameters such as applied target power, N2-gas partial pressure and substrate temperature [9,10,11,29]. A change in these parameters alters the structural and mechanical properties and, hence, the friction resistance of the coating [30]. However, few systematic studies have investigated the effect of pressure on the microstructure and characteristics of CrAgCeN coatings. Moreover, binary doping of RE elements has rarely been reported in the preparation of CrN thin coatings by magnetron sputtering. Therefore, studying the effect of pressure on the structure and composition of coatings is expected to significantly advance the field. In this paper, a CrAgCeN coating was prepared by magnetron sputtering under different pressures. The aim was to elucidate the effect of deposition pressure on the microstructures and tribological properties of the formed coatings.

## 2. Materials and Methods

### 2.1. Coating Deposition

CrAgCeN coatings were prepared by a magnetron sputtering system. The system is composed of a vacuum system, a magnetron sputtering target and a power supply. Three permanent magnetic targets are evenly distributed in the bottom of the vacuum chamber (φ450 × 350 mm) according to 120°, and folded to the center of the sample above. The pre-plated parts are installed on the rotatable workbench above the chamber, and the distance between the parts and the electrode is 85 mm. The power supply has DC cathodes and one RF cathode. The sputtering targets (provided by Beijing commercial company) were Cr (99.95%) and the CeAg alloy (5:5 Ce: Ag at%) (Φ50.8 mm × 3 mm). Ar and N_2_ were used as the working gas and reaction gas, respectively, and 304 stainless steel (Φ30 mm × 2 mm) was selected as the substrate. After evaluating the micromorphologies and wear properties of the CrAgCeN coatings, the hardness *H* and elastic modulus *E* were determined using a single silicon crystal (10 mm × 10 mm × 50 μm). The 304 stainless steel substrates were ground with sandpaper and grinding paste and then polished. Next, the substrates were cleaned with acetone and ethanol for 15 min and placed in a vacuum chamber. A Cr transition layer was deposited for 20 min to improve the adhesion of the coating to the substrate. During a deposition process, both sputtering targets were treated at the same time and the workpiece table was rotated at 20 r/min. The main process parameters were set as follows: vacuum pressure = 5 × 10^−4^ Pa, Ar flow rate = 40 sccm, N_2_ flow rate = 20 sccm, Cr target power = 200 W, CeAg target Power = 60 W, deposition temperature = 200 °C, total coating-deposition time = 120 min, deposition pressure = 0.4, 0.5, 0.6, 0.7, 0.8 Pa.

### 2.2. Coating Characteristics

The surface morphologies, crystal structures, phases, chemical bonding states, average thicknesses, and surface hardness values of the deposited CrAgCeN coatings were characterized by scanning electron microscopy (SEM) (Jeol JSM-7900F, Tokyo, Japan), X-ray photoelectron spectroscopy (XPS) (Thermo Scientific K-Alpha, Waltham, MA, USA), X-ray diffraction (XRD) analysis (Smartlab Rigaku, Tokyo, Japan), and a nanoindentation instrument (iNano, Milpitas, CA, USA).

The tribological performances of the coatings on 304 stainless steel were evaluated in dry friction and sliding wear tests using a reciprocating sliding ball-on-disk tribometer (HT-1000, Lanzhou, China) in ambient air. Counter-friction was provided by a GCr15 ball (Φ6 mm, hardness 63–65 HRC). Tribo-mechanical tests are performed with humidity of 50% ± 5% and temperature of 20 ± 2 °C. The parameters of the ball-on-disk machine were set as follows: speed = 336 r/min, normal load = 3 N, total number of cycles = 3360, rotating radius = 2 mm. The friction coefficient was automatically recorded during a sliding duration of 10 min. The wear-track morphologies on the samples were observed by a white-light interference three-dimensional profiler. The wear area (obtained by integrating the profile) was multiplied by the total length of the wear track to obtain the wear volume.

The wear rate *W* [1 in mm^3^/(N·m)] was calculated as
(1)$$W=\frac{V}{F\times L}$$
where *V* is the worn volume (mm^3^), *F* is the applied normal load (N), and *L* is the total friction stroke (m). The wear rates of three tests were averaged to reduce the error and were used to evaluate the wear performances of the coatings.

## 3. Results

### 3.1. Structural Characterization

Figure 1 shows the surface morphologies of the CrAgCeN coatings. As the deposition pressure increased, the surface topography changed from loose small pyramidal or strip-like structures to denser bars or triangular cones and finally to pyramidal structures. When the sputtering gas was applied at 0.4 Pa, the coating topography was loose and columnar structures were porous (Figure 1a,f). At a higher gas pressure of 0.5 Pa, the porosity of the CrAgCeN coating reduced (Figure 1b,g). The coating grown at 0.6 Pa exhibited a dense topography with compact columnar structures (Figure 1c,h). However, the morphology reverted to loose and lacunose pyramidal structures at deposition pressures of 0.7 and 0.8 Pa (Figure 1d,e) and the columnar structures gradually become less marked (Figure 1i,j). This observation indicates that at appropriate deposition pressures of the chamber, the CrAgCeN coatings became dense and developed a compact columnar structure.

The above results clarify that varying the deposition pressure changes the structure and topography of the CrAgCeN coating. At the initial stage of coating growth, small islands or clusters were formed that served as nucleation sites for subsequent incoming atoms. As these islands or clusters grow, they either coalesce to cover the substrate surface or form individual columns, depending on the mobility of the atoms arriving at the surface [31]. Porous columnar crystals form when the self-shadowing effect of the initial nuclei combines with low lateral diffusion of the adatoms on the substrate surface. According to the kinetic theory of gas molecules, the mean free path (λ) of gas molecules is mainly determined by the temperature (*T*) and pressure (*P*) of the gas molecules; specifically, $\lambda =2.331\times 10^{-20}\left(T/P\delta^{2}\right)$, where δ is the radius of the gas molecules [32,33]. During the deposition procedure, the average energies of the sputtered CrAgCeN particles (atoms and ions) ejected from the target surface are well controlled so the temperature is constant. Therefore, the mean free path of the sputtered CrAgCeN particles is determined only by the pressure of the sputtering gas. Under low-sputtering gas pressures, the mean free path of the sputtered CrAgCeN particles increases and the scattering effect between the background Ar gas and the sputtered CrAgCeN particles is low. The sputtered fluxes can then be considered linear motion. An arriving atom has sufficient energy to move across the surface before coming to rest, so the CrAgCeN coating becomes dense. When the pressure of the sputtering gas reduces further, the mean free path of the sputtered CrAgCeN particles increases but fewer particles are ejected from the target and the particles reaching the surface are sparse [34]; thus, the CrAgCeN coating is loose. In contrast, at high pressures of the sputtering gas, the mean free path decreases and the atoms will likely collide with each other before arriving at the substrate surface. Such collisions reduce the energy of the sputtered atoms at the surface. The lower surface mobility of the atoms results in growth with a porous columnar structure [31,33]. When the pressure of the sputtering gas further increases, the scattering effect between the background Ar gas and the sputtered CrAgCeN particles becomes very high. The average energies of the sputtered atoms are significantly reduced at the surface, greatly lowering the lateral diffusion mobility. Therefore, the columnar structure of the porous CrAgCeN coatings becomes less marked at both high and low pressures, owing to the combined effects of self-shadowing and low lateral diffusion mobility of the adatoms of ejected CrAgCeN particles. As the pressure increases, the energy of Cr, Ce and Ag atoms incorporated with N in the nitride structure decreases because the mean free path in the deposition chamber is reduced and fewer Cr, Ce and Ag atoms form the CrCeAgN structure on the substrate [33]. Additionally, co-doping with Ce and Ag elements significantly reduces the surface energy of the nuclei, increases the number of nucleations, and suppresses the growth of coarse grains [35,36,37].

Table 1 summarizes the chemical compositions of the coatings deposited at different pressures. The Cr/N atomic ratio was approximately 1.2 and maximized at 1.21 under 0.6 Pa pressure. 

Figure 2 presents the XRD spectra of the CrAgCeN coatings deposited at different pressures. All spectra show the characteristic peaks of CrN, Cr_2_N, Ag, AgN_3_ and Ce. The CrN diffraction peaks correspond to the (111), (020) and (200) planes of the face-centered cubic (FCC) phase. These orientations are consistent with ICDD JCPDS card 65-6914 for CrN and the structure reported by Alexander et al. [13]. The Cr and Cr_2_N diffraction peaks correspond to the (111), (211), (100) and (113) planes of the FCC phase, in agreement with ICDD JCPDS cards 35-0803 and 37-1230 for CrN and Cr_2_N, respectively, and with the structure reported by Bílek et al. [38]. The Ag peaks correspond to the (111) and (211) planes of the FCC phase. These orientations agree with ICDD JCPDS card 04-7831 for Ag and the structure reported by Kutschej et al. [15]. The XRD peaks of Ce indicate preferred orientations of (101) and (211). The low intensities of the Ce peaks indicate amorphous-phase Ce, which is beneficial for improving the material properties. As the deposition pressure increased, the intensities of the CrN peaks first decreased and then increased. At a deposition pressure of 0.6 Pa, the (111) and (200) peaks were weakest while the (111) diffraction peak of Cr_2_N remained obvious. Bílek et al. similarly reported only the presence of (111) and (200) FCC orientations in CrN thin coatings [38]. We attribute this effect to the preferential incorporation of Ag on the rough (111) surfaces rather than on the smoother (002) surfaces, which suppresses the continued growth of the 111-oriented column. This inference is consistent with the structure reported by Mulligan et al. [20]. Ce is known to induce the preferred orientation of the TiN phase along a densely packed plane [39]. The AgN_3_ peak was confirmed by XPS but has almost never been reported in previous studies of CrN/Ag coatings. Zhou et al. reported that growth kinetics play an important role in silver nitride formation by nitrogen implantation [40]. Ce has a 4f-orbital electronic structure and strong chemical activity; therefore, it provides strong growth kinetics that might explain the AgN3 peak in the XRD spectrum of CrAgCeN. Scherrer formula [41] was used to calculate the average crystallite size (D)of the samples as represented in Equation (2):(2)$$D=0.9\lambda /\beta \cos\theta_{B}$$
where λ, $\theta_{B}$ and β are the X-ray wavelength (1.54056 Å).The crystallite size as calculated by the Scherrer formula increases from 19 nm to 37 nm with varying deposition pressure. After introducing Ce and Ag, the metal grains were refined, the AgN_3_ peak was enhanced, and the crystallinity increased. The coating deposited at 0.6 Pa contained CrN, Cr_2_N, AgN_3_, Ce and Ag with a tighter grain arrangement and more outstanding comprehensive properties than the coatings deposited at other pressures.

To explore the diffraction pattern of CeO_x_, the bonding state of CeO_x_ was examined by XPS. The N 1s, Cr 2p, Ce 3d and Ag 3d XPS spectra of the coating deposited at 0.6 Pa are shown in Figure 3. The Cr 2p peaks of the CrAgCeN coating represent two phases, Cr_2_N (577.2 eV) and CrN^3+^ (584.8 eV). Similarly, the Cr 2p peaks of CrN coatings represent three phases, and two phases, Cr_2_N (574.1 eV) and CrN (575.0 eV), were analyzed using the XPS results and reference studies [5,17]. The Cr_2_N spectra in Figure 3 confirm the XRD results (Figure 2 at 0.6 Pa). The Ag 3d spectrum of CrAgCeN shows a characteristic peak at a bonding energy of 368.3 eV, consistent with literature reports (368.2 eV). Another characteristic peak at 367.5 eV in the Ag 3d spectrum [42] corresponds to AgN^3+^ in CrAgCeN. The AgN^3+^ spectra of CrN/Ag coatings are almost absent in the literature, possibly because when the Ce element is lacking, the kinetics are insufficient for AgN_3_ formation. The N 1s spectra of the CrAgCeN coating are described by two peaks with different intensities, one centered at 396.7 eV, the other at 399.5 eV. These peaks are consistent with the nitride peaks in the Cr 2p and Ag 3d spectra. The Ce 3d XPS patterns (Figure 3 show three distinct peaks corresponding to Ce^3+^ and Ce^4+^ states, indicating the presence of mixed oxide groups (Ce_2_O_3_ + CeO_2_) on the surface. Ce oxide phases can effectively prevent rapid crystal growth of anatase in Ce-doped TiO_2_ coatings, thus maintaining their ordered structures of CrAgCeN coating due to a full octet of electrons in the 5s^2^p^6^ outer shell [43].

### 3.2. Mechanical Properties

Figure 4 plots the *H*, *E*, *H*/*E* and *H*^3^/*E*^2^ values of the CrAgCeN coatings as functions of deposition pressure. The *H*/*E* and *H*^3^/*E*^2^ values are related to the resistance of elastic strain to failure and plastic deformation, respectively. At a deposition pressure of 0.4 Pa, the *H*, *E*, *H*/*E* and *H*^3^/*E*^2^ values were 10.1, 185.3, 0.055 and 0.03 GPa, respectively. At 0.6 Pa, these values increased to 14.2, 206.2, 0.069 and 0.067 GPa, respectively. These values decreased to 11.7, 186.8, 0.063 and 0.046 GPa, respectively at 0.8 Pa. The hardness of the coatings was enhanced by the fine-grain strengthening of the RE element Ce and the soft Ag phase at 0.6 Pa. The Ce additive increased the mechanical properties of the CrAgCeN coatings. As mentioned in Section 3.1, Ce effectively refined the microstructure, thereby increasing its average hardness [44]. Increasing the amount of soft Ag phase in the coating counteracts the effect of fine-grain strengthening, causing a gradual decline in hardness. At a deposition pressure of 0.6 Pa, the hardness of the coating was maximally improved, indicating that Ce and Ag together improved the hardness.

The *H*, *E*, *H*/*E* and *H*^3^/*E*^2^ values are typically used for evaluating hard and tough coatings. The hardness and toughness of coatings are closely correlated with friction and wear properties [4,13,45]. The *H*/*E* and *H*^3^/*E*^2^ ratios predict the wear resistance of hard coatings; specifically, higher values denote stronger wear resistances [4,7,46]. Under a deposition pressure of 0.6 Pa, the content of Ce and Ag was 3.8% (Table 1) and the *H*/*E* and *H*^3^/*E*^2^ ratios were higher (0.069 and 0.067 GPa, respectively) than at other pressures. Therefore, the coating deposited at 0.6 Pa is expected to show excellent friction and wear properties.

### 3.3. Friction and Wear

Figure 5 shows the coefficient of frictions (COFs) and wear rates of the samples. In a dry-sliding wear test, the COFs of the CrAgCeN coatings were calculated as 0.391–0.609. As the deposition pressure increased, the COF of the films first decreased and then increased. A similar trend was observed for the wear rates of the coatings. The COF and wear rate were maximized (0.609 and 18.9 × 10^−7^ mm^3^/(N·m), respectively) at a deposition pressure of 0.4 Pa because the microstructure of the coating was loose under this pressure. Consequently, the hardness and elastic modulus were small and the wear rate was increased. The COF and wear rate were 0.514 and 6.7 × 10^−7^ mm^3^/(N·m), respectively, at 0.5 Pa. The reduction in COF and wear rate of the coating at 0.5 Pa might be explained by the stronger ion bombardment at higher deposition pressures. When the deposition pressure was increased to 0.6 Pa, the COF and wear rate were reduced to 0.391 and 3.2 × 10^−7^ mm^3^/(N·m), respectively). As the microhardness increases, debris particulates are more easily removed from the contact surface under contact stress and the surface becomes smoother [47]. When the deposition pressure increased to 0.7 Pa, the COF and wear rate increased to 0.429 and 7.8 × 10^−7^ mm^3^/(N·m), respectively. Finally, at 0.8 Pa, the COF and wear rate were 0.463 and 9.9 × 10^−7^ mm^3^/(N·m), respectively. Note that the wear rate was three times higher at 0.8 Pa than at 0.6 Pa. Similar results in CrN were reported by Ren et al. [45]. Reducing the disposition pressure significantly decreased the COF and the wear rate, in other words, improving the wear resistance of the coating [20,48].

### 3.4. Mechanism of the Wear Protection and Morphology of the Worn Surfaces

When studying the tribological properties of coatings, characterizing the wear mechanism is as important as determining the COF and wear volume. To this end, SEM micrographs of the worn surfaces were taken and are displayed in Figure 6. Deformation and damage occurred on all surfaces. Large amounts of wear debris adhered to the worn surfaces during the back-and-forth transfer process. The track width and the *H*/*E* or *H*^3^/*E*^2^ ratio have similar changes, which first decrease and then increase with the increase in sedimentary pressure. The samples deposited at 0.4 and 0.5 Pa were strongly deformed and widely damaged (Figure 6a,b), with outstanding scratches and grooves in the wear tracks. In contrast, the sample deposited at 0.6 Pa was only slightly damaged and the wear tracks were mild, indicating higher wear-resistance performance than at 0.4 and 0.5 Pa (Figure 6c). The samples deposited at 0.7 and 0.8 Pa presented mild scratches and grooves in the wear tracks (Figure 6d,e) but their surfaces were severely deformed and damaged, indicating serious adhesive damage.

As evidenced in Figure 6, the failure mechanism of the CrAgCeN coatings was spalling from the substrate, indicating that adhesive wear dominated the main wear mechanism. The worn particles resulting from the abrasion constitute a third body between the steel and coating surface. Some pitting corrosion areas were also observed on the worn surfaces, which are related to oxidation occurring during wear tests in the atmospheric environment. Recall that as the deposition pressure increased, the COF first decreased and then increased. From the wear results of the CrAgCeN surfaces, we can observe the effects of deposition pressure on the wear mechanism of the CrAgCeN coatings.

Archards’ model predicts a negative correlation between the wear rate and hardness of a material. The Ce additive improves the hardness and strength of the base material, thereby weakening the wear [39,43,49]. The Ag lubricant can also weaken the wear of composite materials [20,21,22,23]. Therefore, the Ce and Ag contents must be appropriately selected to improve both the lubrication property and wear resistance of the basic material. Co-doping of Ce and Ag under a deposition pressure of 0.6 Pa facilitated the formation of the Cr_2_N phase and improved the comprehensive properties of the coatings (as shown Figure 2). The Cr_2_N phase provides the coating with high hardness while the CrN phase confers excellent oxidation resistance and lowers the COF [50]. The hardness of AgN_3_ is about 4.5 Gpa [51]. AgN_3_ decomposes into Ag and N_2_ as the friction time goes on [40]. Ag has low shear strength and acts as a lubricating particle at the interface between the coating and the ball, which can slow down the interaction between the friction pair and the wear scar surface, thus reducing the wear of the coating [21,22]. On a surface with high microhardness, debris particulates are easily removed from the contact surface under contact stress and the surface remains smooth [47].

To further investigate the worn surface and wear debris, the worn surfaces of the counterparts were observed by SEM. Under deposition conditions of 0.4 and 0.8 Pa, the surfaces were seriously damaged and unsuitable for a mechanism analysis. Meanwhile, the coatings deposited at 0.5, 0.6 and 0.7 Pa exhibited similar wear properties, which were convenient for analyzing their wear behavior. Figure 7 shows the SEM images of the GCr15 counter-faces at 0.5–0.7 Pa. When the deposition pressure was 0.5 Pa, the surface of the ball was seriously worn and covered with adhesives. Many deep furrows appeared in the wear marks and a large amount of wear debris appeared on the edges. When the deposition pressure increased to 0.7 Pa, the surface of the pair was seriously worn but the adhesion was small. A large number of shallow furrows appeared in the wear marks and a small amount of debris had accumulated on the edges. When the coating was ground against GCr15, most of the wear occurred on the GCr15 member of the pair. At a deposition pressure of 0.6 Pa, the wear surface of the ball was only slightly worn. The slightly furrowed structure was mainly formed by the peeling of the hard particles from the coating during the wear process, leading to abrasive wear. A small amount of adhesive was also observed. Over time, the wear debris accumulated between the coating and friction pair. Eventually, a small amount of coating was transferred, which was caused by a decrease in the COF. This occurred especially when the Ce and Ag phases were reduced. Under contact stress for a certain time, the concave and convex surface peaks become detached from the coatings and are crushed into fine abrasive. Li and Lui showed that after adding a small amount of Ce (0.1 wt%) to a novel aluminum bronze coating, the microstructure was refined with consequent improvement in the abrasive wear resistance [27]. In our study, the abrasive wear dominated the main wear mechanism at a deposition pressure of 0.6 Pa. Energy dispersive X-ray spectrometry (EDS) confirmed the presence of coating elements on the GCr15 surface. The mass percentages of Ce were obviously lower on the 0.5-Pa and 0.7-Pa counter-faces than on the 0.6-Pa counter-face, implying that smaller quantities of materials were transferred from the 0.5-Pa and 0.7-Pa samples than from the 0.6-Pa sample. Meanwhile, the counter-faces of the 0.5- and 0.7-Pa samples presented low levels of Ce and Ag (Figure 7). Residual Ce and Ag on a surface are known to weaken the friction and wear to some extent during sliding [21,39,49]. This reasoning explains the lower COF and wear rate of the 0.6-Pa sample than of the other samples.

To further analyse the effects of Ce on the wear mechanism of the CrAgCeN coatings, the worn surface of the coating deposited at 0.6 Pa was analyzed by XPS (Figure 8). The N 1s XPS spectrum of the CrAgCeN coating presented two peaks with different intensities, one is centered at 396.7 eV and the other at 399.5 eV. These peaks are consistent with the nitride peaks in the Cr 2p and Ag 3d spectra. The peaks at 583.50 eV (Cr 2p_1/2_) and 574.30 eV (Cr 2p_3/2_) were assigned to Cr [5,17] (Figure 8b). The Ag 3d XPS spectrum of the CrAgCeN coating displayed characteristic peaks at a bonding energy of 368.3 eV, which perfectly agrees with the literature value (368.2 eV) [42] (Figure 8d; recall also Section 3 and Figure 3). Meanwhile, a small amount of Ce was detected on the worn surfaces (Figure 8c). The Ce 3d XPS patterns (Figure 3c) also presented distinct peaks suggesting the formation of mixed oxide groups (Ce_2_O_3_ + CeO_2_) on the surface.

The introduction of CeO_2_ has been shown to improve the oxidative wear resistance of Fe–Ni–Cr alloy coatings [48,49]. The tribological properties of CrAgCeN coatings were enhanced by the addition of Ce, which reduced the COF. This suggestion is supported by the SEM–EDS mappings of worn surfaces on the coatings deposited at 0.5–0.7 Pa. Furthermore, as revealed in the Ce 3d spectrum of the worn coating deposited at 0.6 Pa (Figure 8c), all Ce atoms existed as Ce3d and Ce4d ions. These ions formed mixed oxide groups (Ce_2_O_3_ and CeO_2_) on the worn surfaces. Ceria groups appearing on the worn surface may perform as a solid lubricant that reduces COF [52,53,54]. The lubricant can decrease the interaction forces with the sliding counterpart and contribute to the lowering of COF after the Ce incorporation.

## 4. Conclusions

Coatings consisting of a CrN matrix and an embedded CeAg were co-deposited via reactive magnetron sputtering. The mechanical and tribological properties of the CrAgCeN composite coatings could be improved at the appropriate deposition pressure. The main results are summarized below.

(1)The CrAgCeN coatings presented a FCC structure and the characteristic diffraction peaks of CrN, Cr_2_N, Ag, AgN_3_ and Ce via the XRD analysis.The grain size of the coating first decreased and then increased with increasing deposition pressure.(2)The hardness of the CrAgCeN coating also increased and then decreased with increasing deposition pressure. The maximum hardness (14.2 GPa) was reached at a deposition pressure of 0.6 Pa. Meanwhile, the elastic modulus of the coating gradually decreased with increasing deposition pressure. The hardness of the coating was affected by the strengthening of Ce fine grains and the soft Ag phase. The decrease in the elastic modulus was mainly attributable to the Ag phase, which has an intrinsically low elastic modulus.(3)The average friction coefficient and wear rate of the thin CrAgCeN coatings first decreased and then increased with increasing deposition pressure. The average friction coefficient and wear rate were minimized at a deposition pressure of 0.6 Pa, reaching 0.391 and 3.2 × 10^−7^ mm^3^/(N·m), respectively.(4)The addition of rare earth Ce has better wear resistance than that of the coating with free-Ce. It provides a reference for the preparation of multi-element ceramic matrix coating containing rare earth elements with excellent comprehensive performance. By further adjusting the magnetron sputtering process parameters, the prepared CrAgCeN coating can potentially be used in industrial fields such as machining tools and aeroengine blades.

## Figures and Tables

**Figure 1 materials-16-01141-f001:**
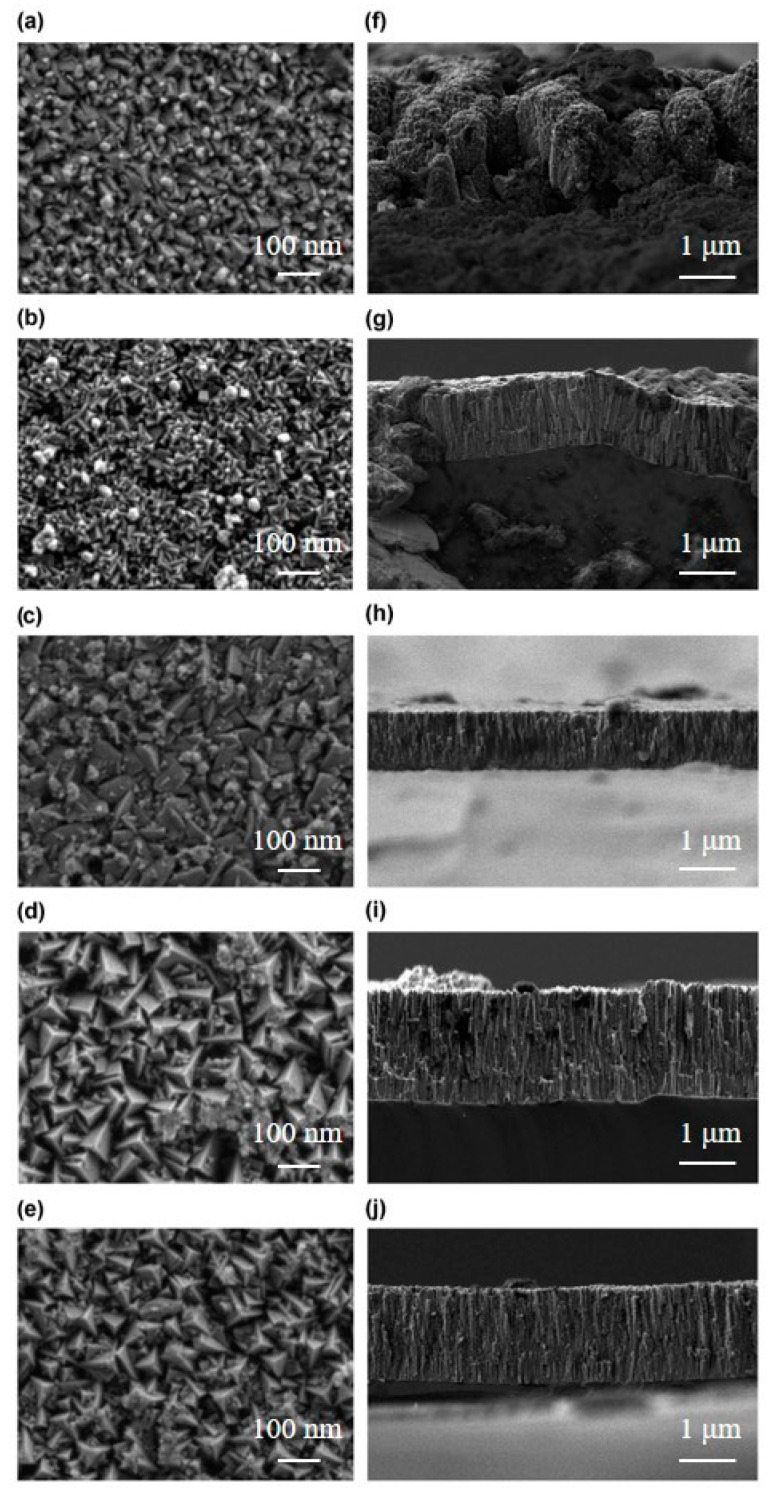
Surface and cross-sectional morphologies of the CrAgCeN coatings formed under different pressures of the sputtering gas: (**a**,**f**) 0.4 Pa, (**b**,**g**) 0.5 Pa, (**c**,**h**) 0.6 Pa, (**d**,**i**) 0.7 Pa and (**e**,**j**) 0.8 Pa.

**Figure 2 materials-16-01141-f002:**
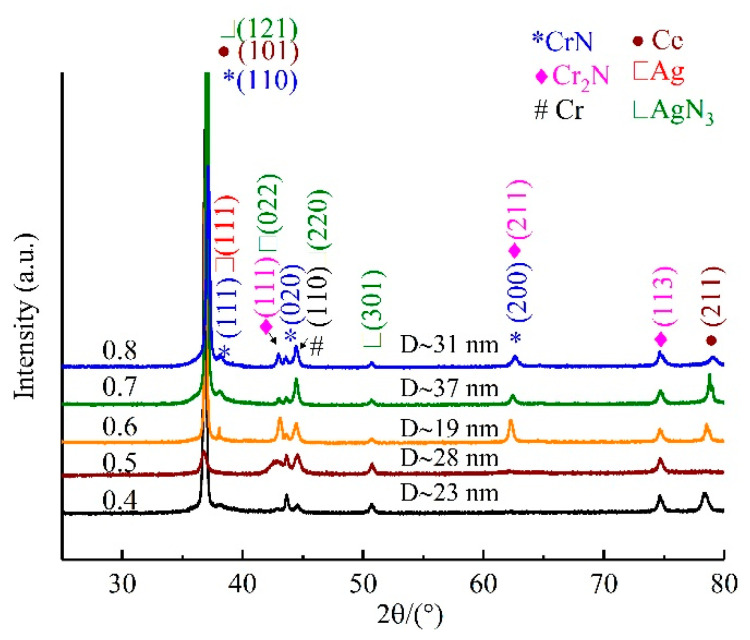
XRD patterns of the CrAgCeN coatings deposited at different pressures (in Pa, listed along the left axis).

**Figure 3 materials-16-01141-f003:**
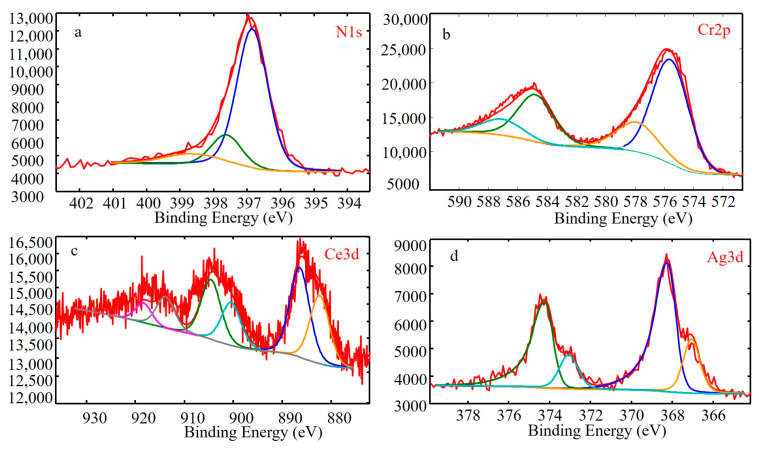
XPS spectra of the CrAgCeN coating deposited at 0.6 Pa (**a**) N1s, (**b**) Cr2p, (**c**) Ce3d, (**d**) Ag3d.

**Figure 4 materials-16-01141-f004:**
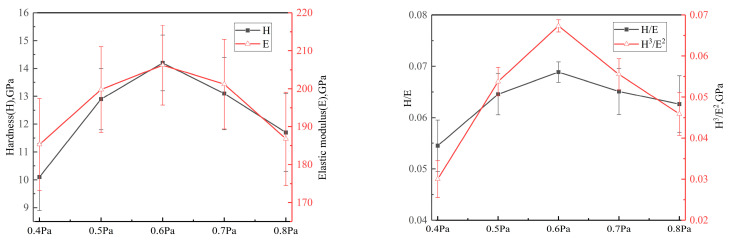
(**Left**) Nanohardness (*H*) and elastic modulus (*E*) and (**right**) *H*/*E* and *H*^3^/*E*^2^ ratios of the CrAgCeN coatings as functions of pressure.

**Figure 5 materials-16-01141-f005:**
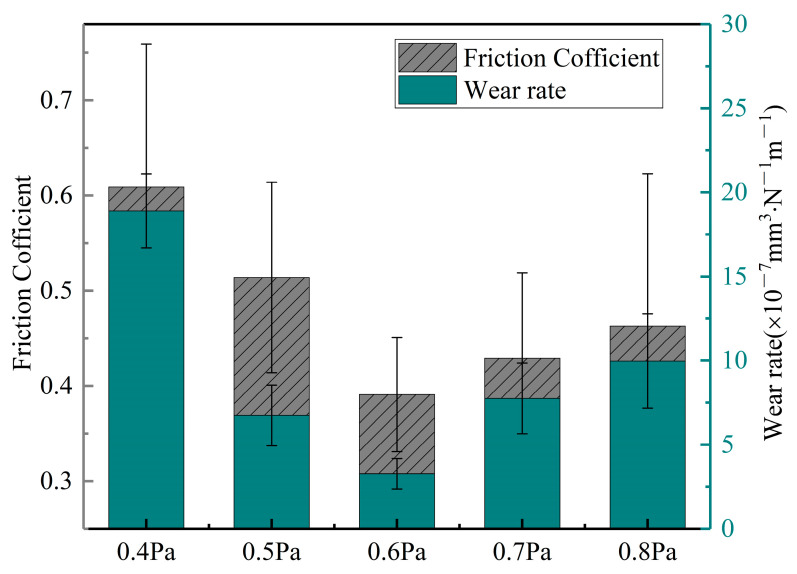
Friction coefficients and wear ratios of the coatings deposited at different pressures.

**Figure 6 materials-16-01141-f006:**
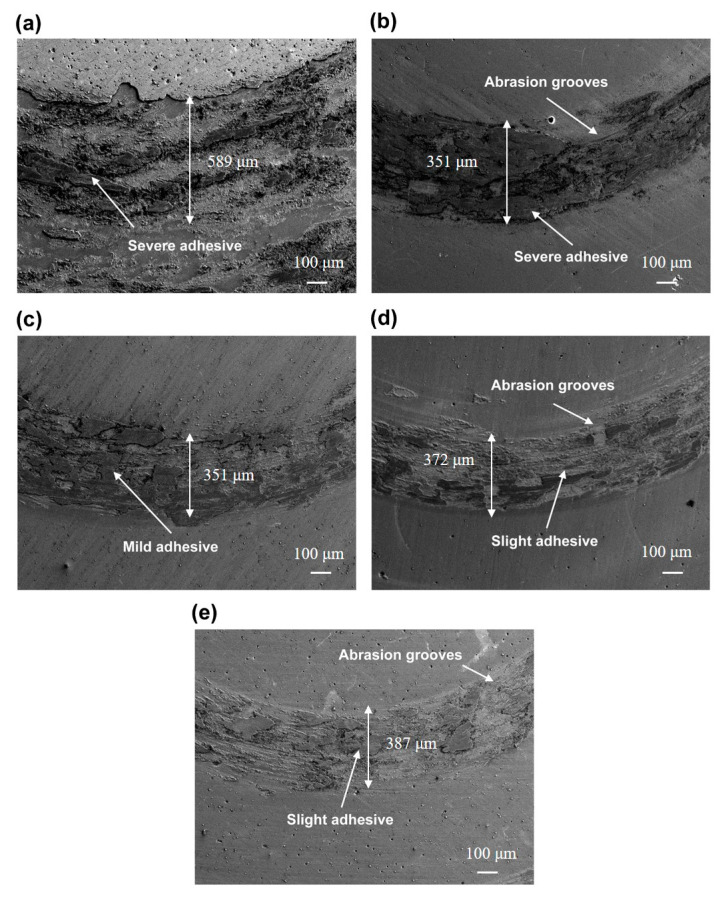
SEM images of worn CrAgCeN coatings deposited at different pressures: (**a**) 0.4 Pa, (**b**) 0.5 Pa, (**c**) 0.6 Pa, (**d**) 0.7 Pa and (**e**) 0.8 Pa.

**Figure 7 materials-16-01141-f007:**
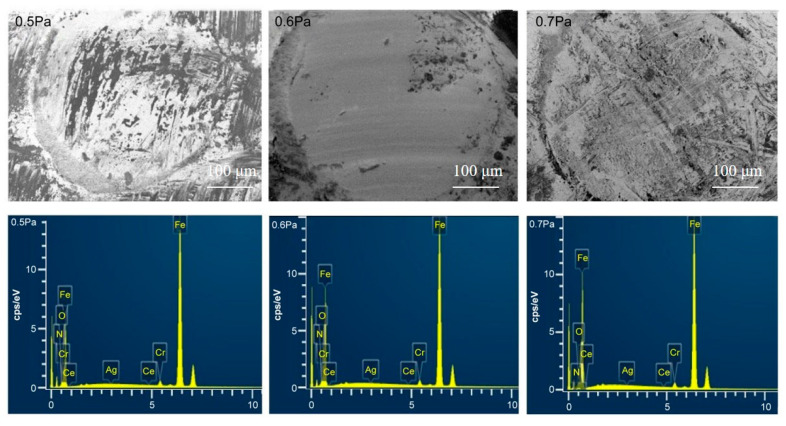
SEM images of the counter-face material (GCr15) after wear testing of coatings deposited at different pressures (left to right: 0.5, 0.6 and 0.7 Pa) under a 3-N normal load.

**Figure 8 materials-16-01141-f008:**
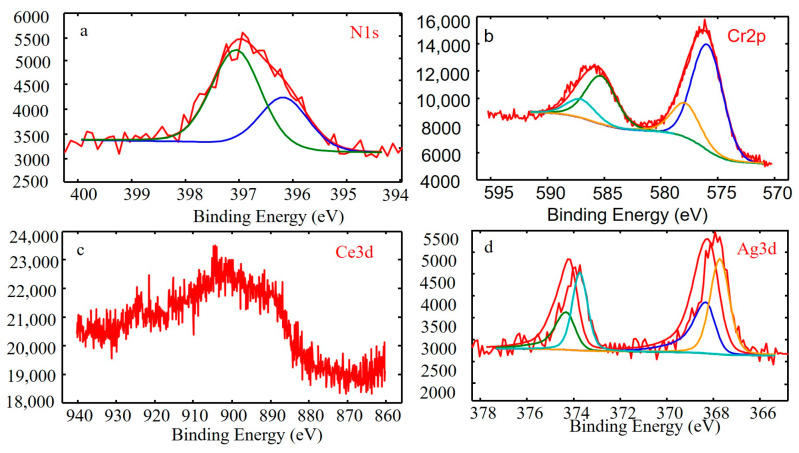
XPS spectra of the worn surface of the CrAgCeN coating deposited at 0.6 Pa (**a**) N1s, (**b**) Cr2p, (**c**) Ce3d, (**d**) Ag3d.

**Table 1 materials-16-01141-t001:** Chemical compositions of the CrAgCeN coatings deposited under different deposition pressure.

Deposition Pressure (Pa)	at.-%Cr	at.-% N	at.-% Ce	at.-% Ag	Cr/N	Ce/Ag
0.4	51.1	44.2	2.5	2.2	1.16	1.14
0.5	51.4	44.5	2.2	1.9	1.16	1.15
0.6	52.6	43.6	2.3	1.5	1.21	1.53
0.7	51.5	44.1	2.2	2.2	1.17	1.00
0.8	50.6	45.1	2.1	2.1	1.12	1.00

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
