# Peer review of "Effects of Deposition Pressure on the Microstructural and Tribological Properties of CrAgCeN Coatings Prepared by Magnetron Sputtering"

_materials, 2023, doi:10.3390/ma16031141_

Round 1

Reviewer 1 Report

Manuscript number: materials-2122339

Title: Effects of Deposition Pressure on the Microstructural and Tribological Properties of CrAgCeN Coatings Prepared by Magnetron Sputtering

The paper presents a study in the field of CrN-based coatings with CeAg obtained during the co-deposition process in MS reactor on steel substrate with the characterization of their selected physiochemical and mechanical properties. The layered materials, especially obtained using MS PVD technique play an important role in modifying of the steel substrate, often to improve mechanical and corrosion behavior, suitable for engineering applications. The methodology is well proposed. Please see my comments below:

ABSTRACT

#1 The abstract presents summarize the research results, please add general information about the novelty and applications of obtained coatings in engineering.

#2 This section is well written but (based on research works from last years) will be suitable to add some information about the role of thin chromium-based coatings (ca. 1 µm) to improve corrosion behavior, as one of the ways during project technology for innovative materials also for application in high temperatures (please see for example 10.17222/mit.2016.151).

MATERIALS AND METHODS:

#3 Section 2.1, lines 65-66: Authors wrote: The sputtering targets were Cr (99.95%) and CeAg alloy (5:5 Ce:Ag) (Φ50.8 mm × 3 mm). Ar and N2 were used as the working gas and reaction gas, respectively, and 304 stainless steel (Φ30 mm × 2 mm) was selected as the substrate.”.

Please in the text the described ratio of “5:5 Ce:Ag”, at. or wag. ratio? Do Autor used a commercial or prepared target material, add some information in the manuscript.

RESULTS

#4 Section 3.1 and 3.4, Figs. 1, 6 and 7: Please add the scale on the presented SEM images.

#5 Figs. 4 and 5: Please add the error value/bars of the presented data, and discuss it (if needed) in the text.

CONCLUSION:

#6 This section presents only the summarized information about obtained measurement parameters value, will be profitable to add a few sentences about the novelty, importance of your research results, and the possibility of applying the proposed technology as functional coatings.

Generally, I recommend the paper for publication in Materials but after minor revision.

Author Response

Dear Reviewers

Thank you very much for your letter and the comments from the referees concerning our manuscript entitled ‘Effects of Deposition Pressure on the Microstructural and Tribological Properties of CrAgCeN Coatings Prepared by Magnetron Sputtering’(materials-2122339). We have revised the manuscript in accord with the comments given by the reviewers. Please check the revised version as reference. We hope that the revision can clarify the questions suggested by the reviewers. Revised portion are highlighted by red in the re-submitted manuscript.We appreciate your consideration of our manuscript, and we look forward to hearing from you soon. Any further information and suggestions are greatly appreciated. Thank you very much for your consideration!

With best regards,

Yours sincerely,

Wei-Hang CHANG

Reviewer 2 Report

The article concerns the PVD (CrAgCeN) coating produced by unbalanced magnetron sputtering. PVD coatings are a widely studied group of coatings. there are several review articles on these coatings. I wonder why there isn't a single review. Considering the subject of the article, I propose to include in the introduction the overview of PVD coatings - Coatings 2020, 10, 921 (doi:10.3390/coatings10100921).

The authors have written: The CrN coating performance is strongly affected by the deposition parameters such as applied target power, N2-gas partial pressure and substrate temperature[8-10. I suggest adding Wear 266 (2009) 800–809, which also presents the effect of various temperatures on CrN coating properties.

line 156: Figure 2 does not show that the coatings are amorphous, please add the size of grains/crystallites based on XRD patterns

lines 191-192:  Add this reference: Coatings 2020, 10, 921 (doi:10.3390/coatings10100921) - there is discussed the effect of H/E and H3/E2 on coating protection properties.

line 193: "higher values denote stronger wear resistances" Yes, but up to a limit value, which is close to H/E = 0.07 or H3/E2 = 0.1 (Coatings 2020, 10, 921, Tribology International 177 (2023) 107991)

Figure 6. Add scale bars to all images.
You can measure the width of the tracks. Is there a correlation between the track width and the H/E or H3/E2 ratio? I wonder if such a correlation exists.

The coatings consist of different phases with different hardness. Did the presence of different phases affect wear? The authors described the influence of the CrN and Cr2N phases on the resistance of the coating, but my comment concerns the degradation mechanisms: does their presence hinder the removal of coating particles? What is the hardness of AgN3 and how this phase affects the destruction process?

line 243: " the coatings deposited at 0.5, 0.6, and 0.7 Pa exhibited similar properties" According to Figure 4, it is not the truth.

The description of the test results indicates the lack of adhesion measurements. These studies should be added. With such a thorough characterization of the coatings produced, the lack of adhesion is evident.

The authors have written that " Energy dispersive X-ray spectrometry (EDS) confirmed the presence of coating elements". I cannot agree. Figure 7 did not show the presence of Ag and Ce peaks. Only the Fe and Cr peaks are distinct

line 273: There is a mistake, there should be Figure 8b instead of 7b

Conclusions:

line 291: "The grain size of the coating first decreased and then increased with increasing deposition pressure." The grain size was not measured, please add it

Author Response

Dear Reviewer:

Thank you very much for your letter and the comments from the referees concerning our manuscript entitled ‘Effects of Deposition Pressure on the Microstructural and Tribological Properties of CrAgCeN Coatings Prepared by Magnetron Sputtering’(materials-2122339). We have revised the manuscript in accord with the comments given by the reviewers. Please check the revised version as reference. We hope that the revision can clarify the questions suggested by the reviewers. Revised portion are highlighted by red in the re-submitted manuscript.We appreciate your consideration of our manuscript, and we look forward to hearing from you soon. Any further information and suggestions are greatly appreciated. Thank you very much for your consideration!

With best regards,

Yours sincerely,

CHANG Wei-Hang

Round 2

Reviewer 2 Report

I have no comments.

Author Response

(The authors gave the same response as above.)
